# Modeling circuit mechanisms of opposing cortical responses to visual flow perturbations

J. Galván Fraile[1]*, Franz Scherr[2], José J. Ramasco[1], Anton Arkhipov[3],
Wolfgang Maass[2], Claudio R. Mirasso[1]

**1** Instituto de Física Interdisciplinar y Sistemas Complejos (IFISC), UIB-CSIC, Palma de Mallorca, Spain,
**2** Institute of Theoretical Computer Science, Graz University of Technology, Graz, Austria, **3** Allen Institute,
Seattle, Washington, United States of America

* jgalvan@ifisc.uib-csic.es

Centre National de la Recherche Scientifique,
FRANCE

**Data Availability Statement:** Code is accessible at:
https://github.com/JavierGalvan9/Billeh_model_
opposing_responses. Data is generated within the

## Abstract

In an ever-changing visual world, animals' survival depends on their ability to perceive and
respond to rapidly changing motion cues. The primary visual cortex (V1) is at the forefront of
this sensory processing, orchestrating neural responses to perturbations in visual flow.
However, the underlying neural mechanisms that lead to distinct cortical responses to such
perturbations remain enigmatic. In this study, our objective was to uncover the neural
dynamics that govern V1 neurons' responses to visual flow perturbations using a biologically
realistic computational model. By subjecting the model to sudden changes in visual input,
we observed opposing cortical responses in excitatory layer 2/3 (L2/3) neurons, namely,
depolarizing and hyperpolarizing responses. We found that this segregation was primarily
driven by the competition between external visual input and recurrent inhibition, particularly
within L2/3 and L4. This division was not observed in excitatory L5/6 neurons, suggesting a
more prominent role for inhibitory mechanisms in the visual processing of the upper cortical
layers. Our findings share similarities with recent experimental studies focusing on the
opposing influence of top-down and bottom-up inputs in the mouse primary visual cortex
during visual flow perturbations.

## Author summary

This study aims to shed light on the intricate dynamics of neural responses within the
mouse primary visual cortex (V1) subjected to visual flow perturbations, unraveling the
emergence of distinct functional classes of excitatory L2/3 neurons, namely depolarizing
(dVf) and hyperpolarizing (hVf) neurons. Through the implementation of a biologically
realistic computational model, the investigation highlights the profound impact of synap-
tic connectivity, inhibitory circuits, and dynamic inputs on shaping these responses. The
identified competition by common inhibition mechanism between dVf and hVf neurons,
driven not only by long-range thalamic inputs but also by local connectivity, provides new
insights into the underlying neural circuitry. This study opens avenues for further explo-
ration into the role of locomotion-related inputs in modulating neural responses, offering

code and Billeh model building files can be accessed at https://portal.brain-map.org/explore/models/mv1-all-layers.

**Funding:** JGF, and CRM acknowledge support from the Spanish Ministerio de Ciencia, Innovación y Universidades, project PID2021-128158NB-C22 funded by the MICIU/10.13039/501100011033. JGF, JJR and CRM acknowledge support from the Spanish Ministerio de Ciencia, Innovación y Universidades, project María de Maeztu CEX2021-001164-M funded by the MICIU/10.13039/501100011033. The contribution of WM and FS was partially supported by the Human Brain Project (Grant Agreement number 785907) of the European Union. The work by AA was supported by the National Institute of Biomedical Imaging and Bioengineering of the National Institutes of Health under Award Number R01EB029813 and the National Institute of Neurological Disorders and Stroke of the National Institutes of Health under Award Numbers R01NS122742 and U24NS124001. The funders did not play any role in the study design, data collection and analysis, decision to publish, or preparation of the manuscript.

**Competing interests:** The authors have declared that no competing interests exist.

a comprehensive framework for future investigations into sensory perception and neural coding.

## Introduction

The brain's proficiency in visual processing, specifically its ability to perceive and respond to dynamic changes in the environment, highlights the intricate mechanisms underlying sensory perception. This capacity is vital for the survival, navigation, and adaptive behaviour of animals in their complex and rapidly changing natural habitats [1–3].

Deciphering the neural basis of motion perception and the response to visual flow perturbations is a crucial objective in sensory neuroscience. The accomplishment of this goal has the potential to shed light into the complexities underlying attentional modulation and decision-making processes [4–6].

In this context, the concept of predictive coding offers a prominent framework to understand how prior expectations influence these responses [7]. It posits that the brain forms internal representations of the world to predict incoming sensory information, constantly striving to minimize the discrepancy between these predictions and actual sensory inputs [8, 9]. This process, which aligns with the principles of Bayesian inference, is central to understanding how the brains interprets and responds to the world. The notion of prediction errors, signaling discrepancies between expected and actual sensory inputs, is a conerstone of this hypothesis. These errors are believed to be computed and relayed by specific neurons to update the brain's internal model, a process observable across various neural circuits including the visual cortex [10, 11], the auditory cortex [12], and the reward system [13].

In the field of visual information processing, particularly within the primary visual cortex (V1), excitatory neurons in layer 2/3 (L2/3) have been identified as key players in responding to changes in visual flow [11, 14, 15]. Intriguingly, these neurons exhibit distinct responses to visual pertubations even in scenarios devoid of explicit sensorimotor expectations [11, 15]. This observation suggests that, despite the well-established role of locomotion in influencing the activity of V1 neurons [11, 14, 16–19], excitatory L2/3 neurons possess the capacity to signal pertubations in visual flow. These signals manifest in two ways: some neurons depolarize, referred to as dVf neurons, while others hyperpolarize, referred to as hVf neurons.

Therefore, an essential research question arises: How does the genetically encoded structure of canonical microcircuits in the neocortex lead to the emergence of these two classes of perturbation-responsive neurons? While some studies have provided useful insights into the computational roles of different neuronal populations [20] and the conditions required for the emergence of prediction error neurons [21, 22], when considering V1 neurons, feature selectivity becomes a prominent aspect, given the role of certain neurons as feature detectors [15, 23–25]. In addition, recent experiments suggest that in the absence of sensorimotor expectation, the response of some neurons to visual perturbations can be solely explained by a combination of neurons feature selectivity and locomotion gain [15]. Therefore, the coexistence of diverse functional types of neurons in V1, including prediction error neurons [11] and perturbation-responsive neurons [15], further underscores the complexity of visual processing in this region of the brain. These types of neurons may share similar characteristics, necessitating a comprehensive understanding of the underlying neural dynamics and circuitry.

The focus of our study aligns with the broader context of predictive coding but focuses on a specific aspect: the emergence of distinct classes of neurons in V1 that are responsive to visual flow perturbations in the absence of sensorimotor expectations. Additionally, we sought to

identify the underlying factors responsible for the emergence of these distinct response classes. Our work encompassed various aspects, including the study of the V1 connectome, the influence of neural and synaptic dynamics, and the interaction between external and internal (recurrent) inputs. Although earlier studies primarily emphasized the structural aspects of brain networks [26, 27], recent evidence highlights the importance of incorporating neural dynamics for a correct description of circuit behavior [28–30]. It is crucial to recognize that the presence of an anatomical connection between neurons does not necessarily guarantee a functional connection, since this also depends on the dynamic activity of the neurons. On the contrary, the existence of functional connectivity does not always imply an underlying anatomical connectivity. In particular, functional hubs, which have been identified as efficient nodes for information transmission in brain networks [31], may not align with structural hubs. The interplay between anatomical and functional connectivity underscores the complexity of brain networks and necessitates extensive research to unravel the mechanisms governing responses to visual flow perturbations. By investigating this phenomenon, we aim to contribute to the understanding of how the brain processes visual information under changing conditions.

In our work, we employed a biologically realistic computational model for the mouse area V1 and the thalamic lateral geniculate nucleus (LGN) [32]. This model, which we refer to as the Billeh model (Figs 1A and 2), integrates a large body of experimental evidence to capture the feature selectivity of V1 neurons in mice. It offers several advantages: (i) it allows us to explore how different configurations of visual input features influence the response of the different V1 cortical layers, enabling comparisons with experimental observations; (ii) it provides direct access to the postsynaptic currents (PSCs) of each neuron, facilitating an in-depth exploration of the role of recurrent and visual inputs in the system's dynamics; and (iii) it offers insights into the contribution of inhibitory neurons to the network dynamics.

In our endeavor to understand the emergence of distinct classes of neurons in response to visual flow perturbations, we conducted extensive numerical simulations using the Billeh model, specifically analyzing the effects of various visual stimuli on excitatory L2/3 neurons. By examining the contributions of the different PSC sources, both excitatory and inhibitory, we sought to unravel the specific roles of each neuron type within the cortical microcircuit. Remarkably, our investigation confirmed that excitatory L2/3 neurons displayed a two-way response to visual flow perturbations (dVf and hVf neurons), in resemblance to recent experimental findings [11, 15]. In particular, this unique behavior was not observed in the infragranular layers (L5/6), which exhibited a one-way response to visual flow perturbations. Through network dynamics analysis, we not only gained insight into the properties of the network that were not directly observed in its structure, but also deepened our understanding of the computational properties of excitatory L2/3 neurons. In particular, we identified the L2/3 circuitry that enables the presence of dVf and hVf neurons within the Billeh model, with inhibitory Parvalbumin neurons playing a key role in this process.

## Materials and methods

### Data-driven cortical laminar microcircuit model

The construction of mammalian cortical microcircuit models, incorporating a vast diversity of genetically, morphologically, and electrophysiologically different neuron types, has been a long-standing challenge due to limited insights into their specific connectivities. Despite these complexities, recent studies [33–35] have culminated in a detailed cortical microcircuit model of the mouse V1 area [32] (Figs 1A and 2), which effectively replicates the computational

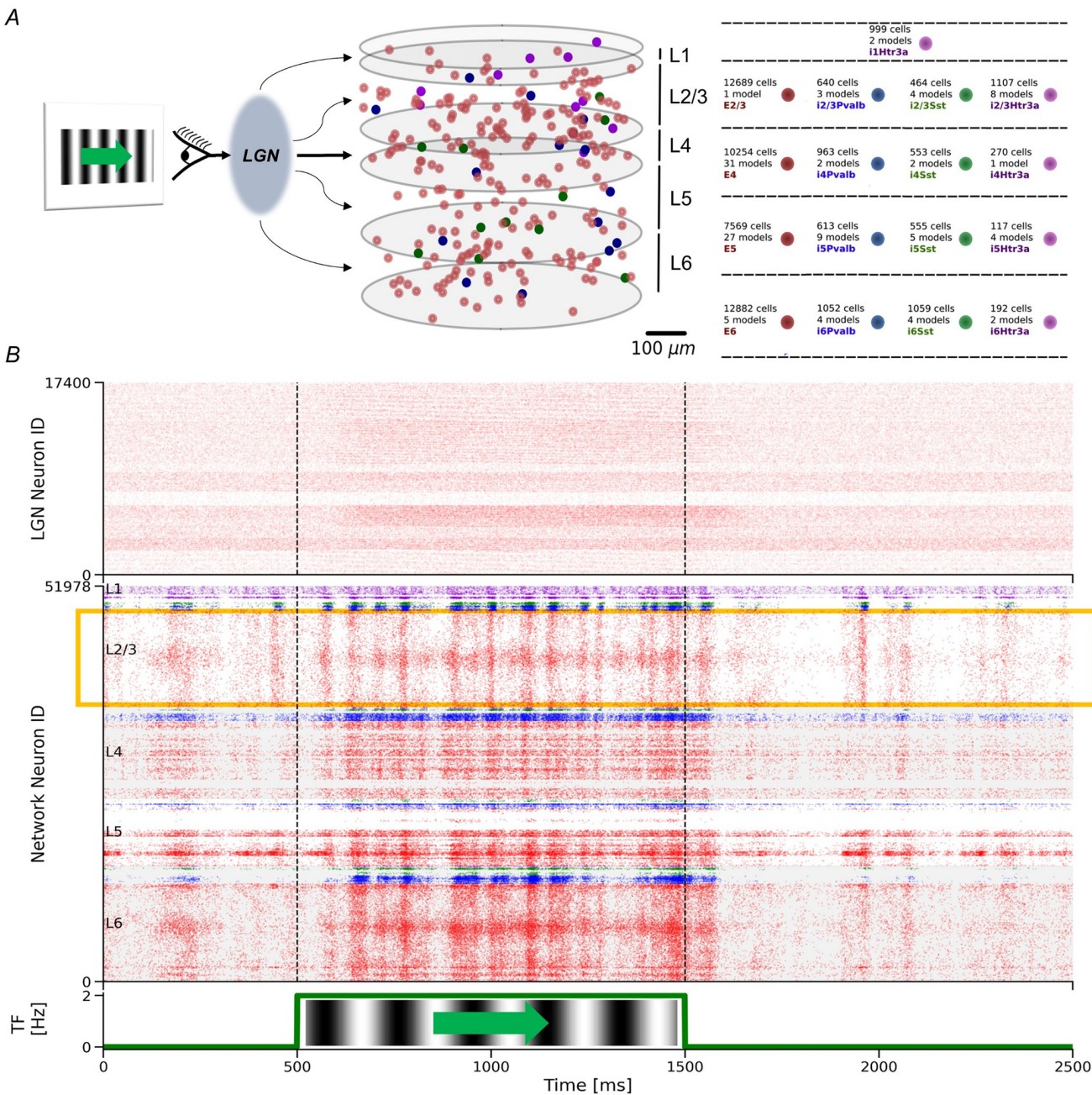

**Fig 1. Billeh model and its response to the sudden onset of visual flow.** (A) The Billeh model describes a patch of mouse cortical area V1 and consists of 230,924 point-neurons divided into one excitatory and three main types of inhibitory neurons (Pvalb, Sst, Htr3a) in each of the cortical layers L2/3, L4, L5, and L6, whereas L1 contains a single inhibitory Htr3a neuron type. Each of these neuron types is subdivided into various models of neurons. The neurons in each population are evenly distributed within a cylindrical domain. The cortical microcircuit model receives the visual input from a thalamus model that transforms the visual input into input currents. In this study, we mainly focused on the detailed "core" region ($400\mu m$ radius) of the model. This visualization shows only 0.5% of those core neurons. (B) Top: Raster plot of the spike response of LGN units to visual stimulus. Middle: Laminar raster plot of the spike response of V1 neurons to the visual stimulus. Different colors represent different populations of neurons, following the same palette as in (A). Several horizontal patches of higher firing rates can be seen due to the directional selectivity of individual neurons. The yellow box highlights excitatory L2/3 neurons, which are the main concern of this research. Vertical dashed lines indicate the period of visual flow. Bottom: Temporal frequency of the visual flow, which consisted of vertical gratings moving horizontally.

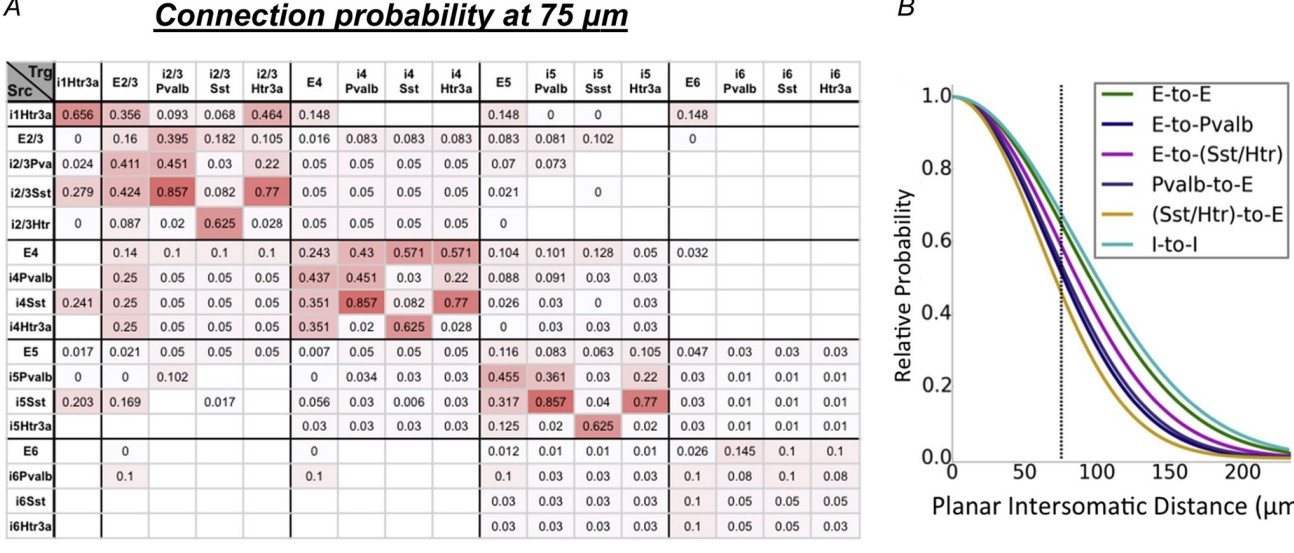

## C Average synaptic weights [pA]

| Source | | L1 iHtr3a | L2/3 E | L2/3 iHtr3a | L2/3 iPvalb | L2/3 iSst | L4 E | L4 iHtr3a | L4 iPvalb | L4 iSst | L5 E | L5 iHtr3a | L5 iPvalb | L5 iSst | L6 E | L6 iHtr3a | L6 iPvalb | L6 iSst |
|---|---|---|---|---|---|---|---|---|---|---|---|---|---|---|---|---|---|---|
| L1 | iHtr3a | -2.24 | -1.70 | -1.25 | -1.10 | -1.64 | -1.15 | 0 | 0 | 0 | -1.54 | 0 | 0 | 0 | -1.07 | 0 | 0 | 0 |
| L2/3 | E | 0 | 0.39 | 13.48 | 22.07 | 15.60 | 1.06 | 4.05 | 12.33 | 8.09 | 2.72 | 0 | 8.83 | 3.49 | 0 | 0 | 0 | 0 |
| | iHtr3a | 0 | -0.90 | -0.60 | -0.40 | -0.90 | -0.80 | -0.53 | -0.51 | -1.15 | 0 | 0 | 0 | 0 | 0 | 0 | 0 | 0 |
| | iPvalb | -0.47 | -1.54 | -0.65 | -1.53 | -1.16 | -1.54 | -0.56 | -1.83 | -1.51 | -0.73 | 0 | -1.60 | 0 | 0 | 0 | 0 | 0 |
| | iSst | -0.60 | -0.99 | -0.82 | -1.13 | -0.42 | -0.82 | -0.72 | -1.36 | -0.55 | -0.81 | 0 | 0 | 0 | 0 | 0 | 0 | 0 |
| L4 | E | 0 | 2.81 | 4.70 | 10.27 | 6.28 | 1.54 | 2.27 | 11.43 | 5.98 | 5.78 | 5.44 | 8.37 | 3.41 | 20.61 | 0 | 0 | 0 |
| | iHtr3a | 0 | -0.93 | -0.58 | -0.40 | -0.90 | -0.79 | -0.54 | -0.50 | -1.17 | 0 | -0.64 | -0.39 | -0.62 | 0 | 0 | 0 | 0 |
| | iPvalb | 0 | -1.79 | -0.66 | -1.54 | -1.21 | -1.75 | -0.56 | -1.87 | -1.51 | -2.69 | -0.73 | -1.92 | -0.82 | 0 | 0 | 0 | 0 |
| | iSst | -0.50 | -0.96 | -0.83 | -1.13 | -0.43 | -0.79 | -0.70 | -1.37 | -0.52 | -1.04 | -0.98 | -0.93 | 0 | 0 | 0 | 0 | 0 |
| L5 | E | 3.19 | 1.50 | 4.70 | 9.25 | 4.71 | 1.04 | 4.05 | 11.04 | 6.09 | 3.02 | 7.88 | 8.05 | 3.41 | 2.55 | 9.15 | 14.55 | 5.37 |
| | iHtr3a | 0 | 0 | 0 | 0 | 0 | -0.80 | -0.49 | -0.49 | -1.20 | -1.03 | -0.56 | -0.33 | -0.67 | -0.70 | -0.84 | -0.32 | -1.05 |
| | iPvalb | 0 | 0 | 0 | -1.17 | 0 | 0 | -0.57 | -2.60 | -1.48 | -2.98 | -0.75 | -2.45 | -0.85 | -2.06 | -0.89 | -2.14 | -1.26 |
| | iSst | -0.40 | -0.80 | 0 | 0 | -1.16 | -0.76 | -0.68 | -1.20 | -0.96 | -0.99 | -0.97 | -0.82 | -0.77 | -0.69 | -1.09 | -0.73 | -1.28 |
| L6 | E | 0 | 0 | 0 | 0 | 0 | 0 | 0 | 0 | 0 | 0.85 | 7.71 | 16.68 | 3.41 | 6.74 | 14.67 | 19.86 | 8.55 |
| | iHtr3a | 0 | 0 | 0 | 0 | 0 | 0 | 0 | 0 | 0 | -1.02 | -0.66 | -0.36 | -0.69 | -0.71 | -0.71 | -0.32 | -1.05 |
| | iPvalb | 0 | -0.26 | 0 | 0 | 0 | -0.22 | 0 | 0 | 0 | -0.30 | -0.78 | -2.46 | -0.83 | -0.21 | -0.90 | -2.12 | -1.31 |
| | iSst | 0 | 0 | 0 | 0 | 0 | 0 | 0 | 0 | 0 | -0.99 | -0.90 | -0.82 | -0.85 | -0.69 | -1.13 | -0.72 | -1.26 |

**Fig 2. Overview of the data-driven cortical laminar microcircuit model of Billeh et al (2020) [32].** (A) Base connection probabilities at $75\mu m$ intersomatic distance depending on presynaptic (Src) and postsynaptic (Trg) neuron types. White grid cells indicate unknown values, which are assumed to be zero in the model. (B) Gaussian scaling of the base connection probabilities shown in (A) as a function of the intersomatic distance for the different types of connections. The probability of a synaptic connection is obtained by multiplying the base connection probability for the presynaptic and postsynaptic neurons with this scaling function. (C) Average synaptic weights depending on presynaptic (Source) and postsynaptic (Target) neuron types.

properties of real V1 circuits. Additionally, the model incorporates an LGN module, designed to simulate thalamic action potentials in response to visual stimuli (Fig 1A).

**Structure of the model.** We implemented the point-neuron version of this model to explore the network dynamics. The model represents a V1 patch with a radius of 845$\mu m$, encompassing 230,924 point neurons. Within this, 51,978 neurons were located in the "core region", a 400$\mu m$ radius internal cylinder surrounded by an annulus to mitigate border artifacts. Neurons were categorized into various types, including excitatory, Htr3a-positive (Htr3a) inhibitory, Parvalbumin (Pvalb) inhibitory, and somatostatin (Sst) inhibitory neurons, spread across layers L2/3, L4, L5, and L6. Besides, L1 exclusively contained Htr3a inhibitory neurons (Fig 1A). This classification resulted in 17 data-based neuron types, characterized by 111 generalized leaky integrate-and-fire models, each derived from the Allen Cell Types Database [36]. Notably, the model considered a single L2/3 excitatory neuron model, implying that unique dynamic behaviors in these neurons arise from differences in neuron input [32].

**Model connectome.** The model's connectome is based on experimental data and Gaussian scaling relative to planar intersomatic distances [32] (Fig 2A and 2B). Postsynaptic weights were determined considering planar intersomatic distance, retinotopic visual space distance, and functional rules based on preferred direction angle similarity. This led to the synaptic weights shown in Fig 2C. Important functional rules include:

1. E-E synapses exhibit "like-to-like" preferences, with neurons sharing similar motion directions being preferentially and more strongly connected [32, 37]. Similar connectivity rules apply to the strength (but not the probability) of other pairs of neuron types.

2. A decrease in E-E synaptic strength with increasing retinotopic visual space distance projected to the target neuron preferred direction.

3. Retinotopic correction in E-E synaptic weights to prevent asymmetric retinotopic magnification.

The network comprises 70,139,111 synapses, with a 0.263% connection probability between two random neurons.

**Neuron models.** The generalized leaky integrate-and-fire model with after-spike currents ($GLIF_3$), a variant of the conventional leaky integrate-and-fire (LIF) model, includes neuronal refractoriness by incorporating slow ion-channel effects with a set of two after-spike currents (ASC), $I_j^m$, with slow and fast time scales. The membrane potential $v_j(t)$ and spiking dynamics $z_j(t)$ are defined by

$$
\begin{aligned}
v_j(t + \delta t) &= \alpha v_j(t) + \frac{1-\alpha}{C}\tau\Big(I_j^{ext}(t) + I_j^{int}(t) + gE_\mathrm{L}\Big) \\
I_j^{ext}(t) &= I_j^{in}(t) + I_j^{noise}(t) + I_j^{rec}(t) \\
I_j^{int}(t) &= \sum_m I_j^m(t) \\
z_j(t) &= H(v_j(t) - v_\mathrm{th})
\end{aligned}
\tag{1}
$$

where $C$ is the neuron capacitance, $g$ is the membrane conductance, $E_L$ is the resting membrane potential, $I_j^{ext}$ is the postsynaptic current, $I_j^{int}$ is the total contribution of ASC, and $v_\mathrm{th}$ the firing threshold of the membrane potential. The decay factor $\alpha$ in the model is defined as $e^{-\delta t/\tau}$, where $\tau = C/g$ is the membrane time constant, and $\delta t$ is the discrete-time step size, set to 1$ms$ in our simulations. $H$ denotes the Heaviside step function.

Each after-spike current $I_j^m(t)$ in the model follows its own dynamic equation, characterized by predefined time scales, $k_j^m$, and an after-spike additive constant, $A_j^m$, as follows

$$I_j^m(t + \delta t) = e^{-k_j^m \delta t} I_j^m(t) + z_j(t) A_j^m; \quad m = 1, 2 \tag{2}$$

The neurons fires when $z_j(t) = 1$, after which they enter in a short refractory period, $\delta_r$, that, depending on the neuron type, ranges from 2$ms$ to 8$ms$. Following this period, the neuron's state variables are updated according to the following reset rules

$$\begin{aligned} v_j(t_+) &\leftarrow v_{r,j} \\ I_j^m(t_+) &\leftarrow R_j^m \times I_j^m(t_-) + A_j^m; \quad m = 1, 2 \end{aligned} \tag{3}$$

with $v_{r,j}$ the membrane voltage reset value, $R_j$ a multiplicative constant, which is typically set to 1, and $t_+$ and $t_-$ representing the time just after and before a spike, respectively.

This neuron model parameters were fit to maximize the likelihood of the simulated neuron reproducing the spike train observed in real neurons [38].

**Synaptic dynamics.**   For each neuron $j$, each postsynaptic current source, $I_j^{\mathrm{syn}}$, was defined by alpha function dynamics with governing equations

$$\begin{cases} I_j^{\mathrm{syn}}(t + \delta t) = e^{-\frac{\delta t}{\tau_{\mathrm{syn}}}} I_j^{\mathrm{syn}}(t) + \delta t e^{-\frac{\delta t}{\tau_{\mathrm{syn}}}} C_j^{\mathrm{syn}}(t) \\ C_j^{\mathrm{syn}}(t + \delta t) = e^{-\frac{\delta t}{\tau_{\mathrm{syn}}}} C_j^{\mathrm{syn}}(t) + \sum_i W_{ji}^{\mathrm{syn}} z_i(t) \frac{e}{\tau_{\mathrm{syn}}} \end{cases} ; \quad \mathrm{syn} \in \{\mathrm{in, \ noise, \ rec}\}. \tag{4}$$

These equations incorporate the synaptic time constant, $\tau_{syn}$, the influence of presynaptic spikes, $z_i(t)$, which could be an LGN unit (*in*), a Poisson source (*noise*) or another V1 neuron (*rec*), and the synaptic weights $W_{ji}^{\mathrm{syn}}$.

The synaptic time constants $\tau_{syn}$ took values 5.5$ms$ for excitatory-to-excitatory (E-E) synapses, 8.5$ms$ for inhibitory-to-excitatory (I-E) synapses, 2.8$ms$ for excitatory-to-inhibitory (E-I) synapses, and 5.8$ms$ for inhibitory-to-inhibitory (I-I) synapses. Additionally, the synaptic delays, arising from the physical distance between neurons, were distributed within a range of 1 to 4$ms$ This distribution was derived from empirical data presented in [32] and adapted to the model's discrete time steps.

**LGN model.**   The thalamic lateral geniculate nucleus (LGN) is known to project retinal inputs to V1 via excitatory projections, which represents a bottom-up input [39]. The Billeh LGN model comprises 17,400 LGN units, each representing a spatio-temporally separable filter. These units process visual stimuli in the form of gray images or movies, outputting time series of instantaneous firing rates. These rates are then converted into spike trains using a Poisson process.

The diversity of the mouse LGN is incorporated by sampling these units from 14 subclasses. The innervation pattern of these units is selective; they predominantly target excitatory and Pvalb neurons across layers L2/3 to L6, and Htr3a neurons in L1, aligning with empirical findings on LGN projections in the mouse brain [20].

**Noise model.**   To simulate the background activity (BKG) typically present in neural circuits, a noise model is incorporated. This model consists of a single Poisson source, firing at 1$kHz$, which injects random excitation into all V1 neurons. The synaptic weights for this BKG noise varied depending on the target neuron type.

**Initial conditions.**   All model state variables were initialized to zero at the beginning of the first simulation trial.

## Visual stimuli

**Drifting gratings.**   The visual stimuli used in the simulations closely resembled those employed in predictive coding experiments [11, 15]. These stimuli consisted of full-field sinusoidal gratings defined by a spatial frequency of 0.04 cycles/˚ and a contrast level of 0.8. These gratings were set to move uniformly accross various directions, ranging from 0˚ to 315˚ in 45˚ increments. In this context, a 0˚ orientation corresponds to vertically oriented gratings moving rightward, while 90˚ represents horizontally oriented gratings moving downward.

Each simulation began with a 500*ms* static grating display. This was followed by a sudden shift to a drifting grating for 1000*ms*. After this dynamic phase, the grating returned to a static state for another 1000*ms*.

**Full field flashes.**   In the latter part of our study, we introduced a sequence of full-field flashes. This consisted of displaying a full-field black image for 500*ms*, followed by a full-field white image for 1000*ms*. This sequence concluded with the display of another black image for an additional 1000*ms*.

## Analysis of the results

For each simulation, we recorded spikes, membrane potential, and input currents for every neuron at each integration step. However, the utility of membrane potential data in understanding network dynamics was somewhat limited, primarily due to the pronounced reset following a spike. Additionally, whereas dopaminergic neurons have the capability to signal bidirectional variations in their inputs [40], the spiking activity of V1 excitatory L2/3 neurons is limited to unidirectional signaling, specifically indicating an increase in their input. This limitation stemmed from their inherently low spontaneous firing rates [41, 42], thereby obscuring aspects of the depolarizing and hyperpolarizing responses in these neurons.

Despite these challenges, we captured the dynamics of the firing rates by sampling them at 60 *Hz* and applying a 150 *ms* Gaussian filter for smoothing. Upon detailed analysis, we determined that the most appropiate parameter for characterizing the network dynamics was the total postsynaptic current, or simply total input current, injected into individual neurons given by

$$I_j(t) = I_j^{ext}(t) + I_j^{int}(t) \tag{5}$$

Baseline values were established during the 500 *ms* period preceding the onset of visual flow. The network's response to the onset of visual flow was then obtained by subtracting these baseline values, enabling a clearer understanding of the network's reaction dynamics.

**Determination of the rheobase.**   For each cell model, we determined its *rheobase*, denoted as $\theta_{rheo}$, which represents the minimum injected somatic current required to elicit a spike in response to rectangular-shaped current stimulation. we applied rectangular current pulses for 1000*ms* to the cell model, starting with a 1*pA* pulse and incrementing by 0.01*pA* after every 1*s* resting interval, until a spike was elicited. The current at which the cell first fired was recorded as its rheobase (illustrated in S1 Fig).

**Classification of excitatory L2/3 neurons.**   Neurons were classified based on their average response to visual flow, determined as

$$\Delta I_j = <I_j> - I_{j,0} = <I_j^{in}> + <I_j^{noise}> + <I_j^{rec}> + <\sum_j I_j^m> - I_{j,0}, \tag{6}$$

where $I_{j,0}$ is the baseline value of the total input current, $I_j^{in}$ is the LGN bottom-up current, $I_j^{noise}$

is the BKG noise, $I_j^{rec}$ is the PSC due to other V1 neurons, and $I_j^m$ represent the $m$-th ASC. Averages were taken over 20 trials and 1000$ms$ duration.

Those neurons exhibiting an increase in their total input current during the visual flow, $\Delta I_{in}$, that exceeded a certain threshold, $\theta_d$ were identified as depolarized with the visual flow (dVf). Conversely, neurons whose total input current response was lower than a certain value, $\theta_h$, were classified as hyperpolarized with visual flow (hVf). Neurons not meeting these criteria were labeled as unclassified.

Since each neuron type had individual characteristics, thresholds were set at ±0.05 times the cell's rheobase. Despite the arbitrary nature of these thresholds, our results proved robust across a range of threshold values.

$$\begin{cases} \Delta I_j & < -0.05 \times \theta_{rheo} & \rightarrow \textbf{hVf} \\ \Delta I_j & > +0.05 \times \theta_{rheo} & \rightarrow \textbf{dVf} \\ & other & \rightarrow \textbf{unclassified} \end{cases} \qquad (7)$$

**Ripley's *K* function.**   To investigate the spatial distribution homogeneity of excitatory L2/3 neurons and to explore the tendency of neurons of the same class, i.e. dVf, hVf or unclassified, to form clusters, we determined the Ripley's *K* function for each class [43], which is defined as

$$K(t) = \frac{A}{n^2} \sum_{i=1}^n N_i(t), \qquad (8)$$

where $A$ is the circular area of layer 2/3, $t$ is the search radius, the index $i$ runs over the cells in the class, and $n$ is the total number of cells in the class. $N_i(t)$ represents the number of cells in the class within an intersomatic planar distance $t$ from cell $i$. When the search area falls partially outside of the V1 column, we apply a weighting factor based on the ratio of the search area that falls within V1 to the total search area. The effects of edge corrections are more important for large $t$ because large search circles are more likely to be outside the V1 column, leading to an underestimation of Ripley's K real value.

This statistical measure evaluates spatial clustering compared to a null model of random homogeneous distribution, which scales as $k(t) = \pi t^2$ until the system edge is reached. If the $K(t)$ value exceeds the random curve for a certain search radius $t$, it indicates spatial clustering at that scale.

**Effective synaptic weight.**   We defined the *effective synaptic weight* between a presynaptic neuron $i$ and a postsynaptic neuron $j$ as

$$W_{neuron\ i \rightarrow j}^{eff} = \frac{1}{\Delta t} \sum_{\Delta t} W_{ij}^{rec} x_i(t), \qquad (9)$$

taking into account the recurrent synaptic weight, $W_{ij}^{rec}$, and the firing state of the presynaptic neuron, $x_i(t)$, which takes the value 1 if at time $t$ the presynaptic neuron fired and 0 otherwise, over a period $\Delta t$. To determine the *effective synaptic weight* from neuron type $i$ (presynaptic) to neuron $j$ (postsynaptic), we summed the contributions from each presynaptic neuron of type $i$,

$$W_{type\ i \rightarrow neuron\ j}^{eff} = \sum_{i=1}^{N_i} W_{neuron\ i \rightarrow j}^{eff}, \qquad (10)$$

where $N_i$ represent the number of type $i$ neurons.

## Statistics

For statistical analysis, each simulation included 20 trials, although the first trial was excluded, that were averaged when computing the desired quantities. To compare mean values among more than two classes (e.g., hVf, dVf, and unclassified L2/3 neurons), we initially conducted a one-way ANOVA test. In case of a significant result ($p < 0.01$), we further applied two-sided Welch's t-tests to assess the significance of differences between the mean values of these groups. The significance levels were denoted as follows: ns (not sifnificant), * ($p < 0.05$), ** ($p < 0.01$), *** ($p < 0.001$), **** ($p < 0.0001$).

In those cases where only two distributions were compared (e.g., excitatory L2/3 versus L5/6 neurons), a two-sided Welch's t-test was directly performed.

For graphical representation of the data, we sometimes employed box-and-whisker plots. In these plots, the box represents the interquartile range and the median, while the whiskers extend to the $10^{th}$ and $90^{th}$ percentiles.

## Results

### Excitatory L2/3 neurons exhibit opposing responses to visual flow perturbations

In the context of visual flow perturbations, divergent definitions have been adopted in experimental studies, considering either sudden visual flow onset or halt [11, 15, 44]. In this investigation, we focused predominantly on sudden visual flow onset, while also briefly exploring the comparison between sudden onset and sudden halt scenarios. Thus, we initiated our study by exposing the Billeh model to vertically static gratings that suddenly started drifting horizontally at a frequency of 2 Hz (see Methods for details and Fig 1B).

Consistent with the bidirectional mismatch responses detected in previous experimental studies [11, 15], we observed excitatory L2/3 neurons with depolarizing and hyperpolarizing responses at the onset of visual flow (Fig 3A). To classify these responses, we labeled neurons as depolarized (dVf) or hyperpolarized (hVf) with visual flow based on whether their average input current change at visual flow onset surpassed 0.05× the rheobase current (dVf) or fell below −0.05× the rheobase current (hVf), the rheobase current being the threshold above which a neuron fires (see Methods for details and S1 Fig). Additionally, excitatory L2/3 neurons that did not show a clear response to the stimulus were identified as **unclassified** (unc) (see S2 Fig).

Before visual flow onset, all classes of excitatory L2/3 neurons exhibited similar values for the different input current sources, maintaining an excitatory-inhibitory balance among recurrent currents with minimal recurrent contribution compared to the LGN current (S1 Table).

Upon visual flow onset, significant changes were observed between excitatory L2/3 classes. Depolarization in dVf neurons primarily stemmed from increased LGN currents, while recurrent inputs remained relatively stable (Fig 3A and S1 Table). Conversely, hyperpolarization in hVf neurons may have resulted from reduced recurrent excitation, possibly due to increased activity of inhibitory presynaptic neurons, reduced activity of excitatory presynaptic neurons, or a combination of both. Notably, there were slightly more neurons in the hVf class compared to the dVf class (dVf, 23.2% (2948/12689); hVf, 26.5% (3363/12689), unclassified, 50.3% (6378/12689)) (S1 Table).

Interestingly, the total input current (Fig 3A) and firing rate (Fig 3B) of dVf and hVf neurons exhibited opposite phases, so that hVf neurons were suppressed when dVf neurons became more active, and vice versa. Baseline neural behaviors also differed significantly between excitatory L2/3 classes, as reflected in the mean and standard deviation of baseline

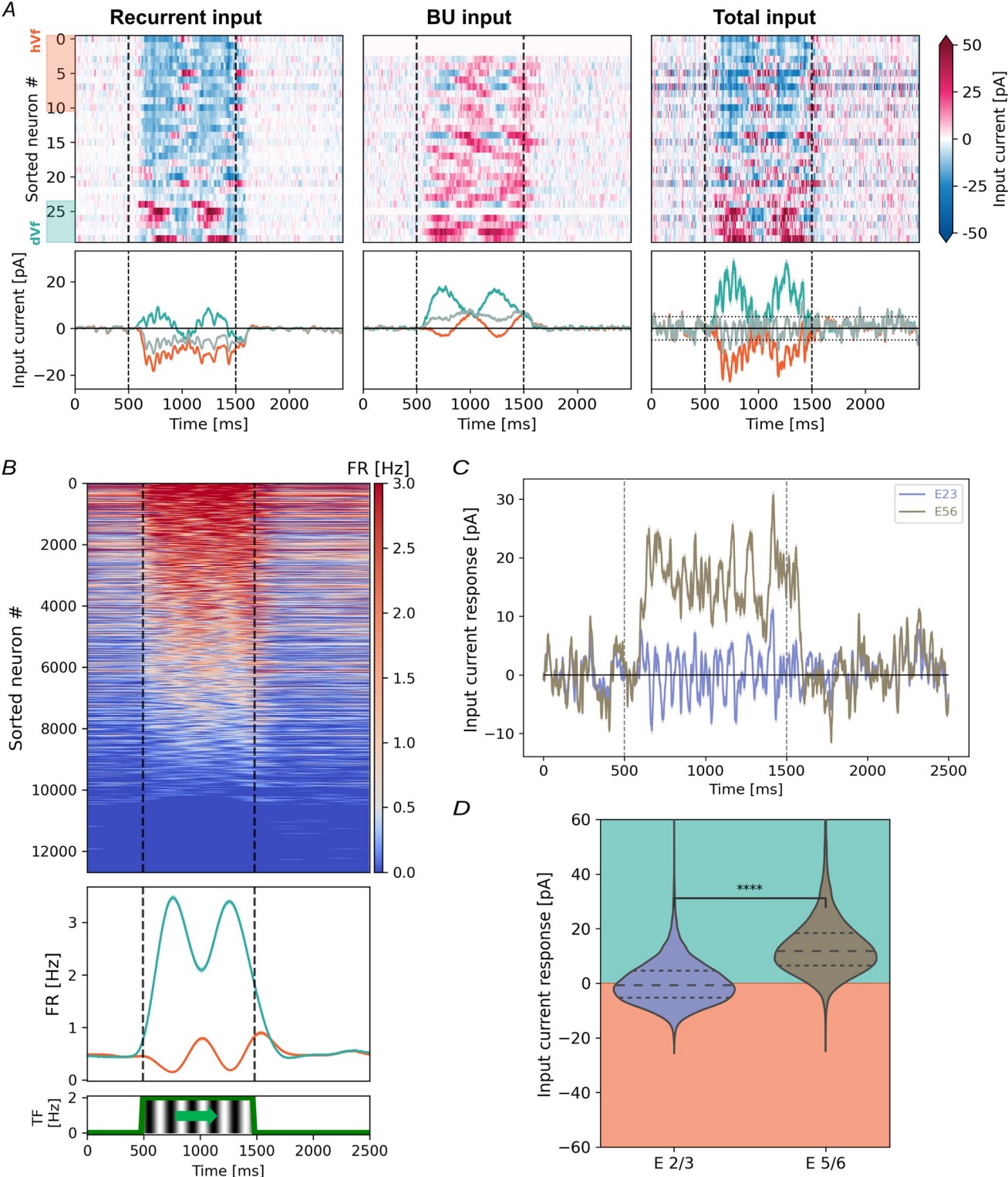

**Fig 3. Opposing excitatory L2/3 neural responses to visual flow onset.** (A) Change in excitatory L2/3 neurons PSCs to 2Hz spatial frequency visual flow onset, accros 20 trials. Include inputs from V1 neurons (left) and LGN (middle). Also, the total PSC is shown (right). Top: Heatmap of 30 randomly selected neurons, sorted by average response. White rows show zero input. Bottom: Class traces with SEM shading. Horizontal dotted lines mark classification thresholds. (B) Top: Firing rates (FR) for excitatory L2/3 neurons. Middle: FR averaged for dVf (turquoise) and hVf (orange) neurons. Bottom: Visual stimulus temporal frequency. (C) Input current response of excitatory L2/3 and L5/6 neurons to visual flow onset. Shading indicates the

SEM across neurons and realizations. (D) Average input current response of excitatory L2/3 and L5/6 neurons to visual flow onset ($t = -110.7$, $p = 0$, $n = 33140$, $n_{E2/3} = 12689$, $n_{E5/6} = 20451$, Welch's t-test). Turquoise and orange shading indicate depolarization and hyperpolarization regions. Horizontal lines denote quartiles of the distributions.

input distributions (S3(A) and S3(B) Fig). This correlation between characteristic visual flow onset responses and baseline behaviors parallels experimental findings [11].

## Divergent visual integration strategies in excitatory L2/3 and L5/6 neurons

Prior research has proposed that the infragranular layers, namely L5/6, employ different visual input integration mechanisms compared to L2/3 [8, 20]. In particular, the sudden onset of drifting gratings in open-loop experiments triggered widespread depolarization among excitatory L5/6 neurons [11].

Our findings obtained using the Billeh model in the absence of top-down input also revealed widespread depolarization of excitatory L5/6 neurons upon visual flow onset (Fig 3C and 3D and S1 Table), consistent with a positive integration of visual input within this layer. Specifically, excitatory L5/6 neurons displayed a marked imbalance between depolarizing and hyperpolarizing responses (dVf: 62.7% (12816/20451); hVf: 0.3% (67/20451); unclassified: 37% (7568/20451)). Consequently, a significant difference was observed between average responses to visual flow onset in excitatory L2/3 and L5/6 neurons (mean ± SD, L2/3: 0.2 ± 7.9 pA; L5/6: 13 ± 11 pA) (Fig 3D).

## Connectome exploration of excitatory L2/3 neuron classes

The observed division of excitatory L2/3 neurons into distinct classes in the sole presence of a visual input that varies over time encourages an exploration of the underlying fundamental principles driving this phenomenon. Computational studies on cortical microcircuits have shown that prediction error neurons may require a balance between excitation and inhibition in multiple pathways, highlighting the relevance of network connectivity [22].

At first glance, it would be reasonable to assume that there exists a difference in the connectivity of the hVf and dVf neurons, i.e., that they were affected by significantly different synapses. Notably, we found that the main difference lay in the number of connections from the LGN, with dVf neurons receiving almost twice as many as hVf neurons did (S2 and S3 Tables). In fact, many hVf neurons lack direct connections from LGN, indicated by zero in-degree values, i.e. number of incoming connections to a neuron (Fig 4A). This observation aligns with the existing idea that the LGN predominantly projects to L4, with projections to L2/3 being sparse and exhibiting weaker connectivity [45, 46].

The analysis of the probability of connection between pairs of excitatory L2/3 neurons displayed a tendency indicating that neurons of the same class, such as two dVf or hVf neurons, were more likely to be connected to each other than to neurons of a different class (Fig 4B). Interestingly, the strength of connections between neurons of the same class was stronger compared to connections between neurons of different classes (Fig 4C).

This pattern closely aligns with a *modular structure* in dVf and hVf neurons, a feature observed in cortical functional networks [29, 47–49]. In a modular structure, neurons within the same module (or class) are more densely interconnected, which facilitates specialized processing within the cortical column [45, 50]. Similar patterns have been identified in fosGFP-expressing layer 2/3 pyramidal cells in the primary somatosensory cortex, which displayed elevated spontaneous activity and were more likely to form connections with each other [45, 51].

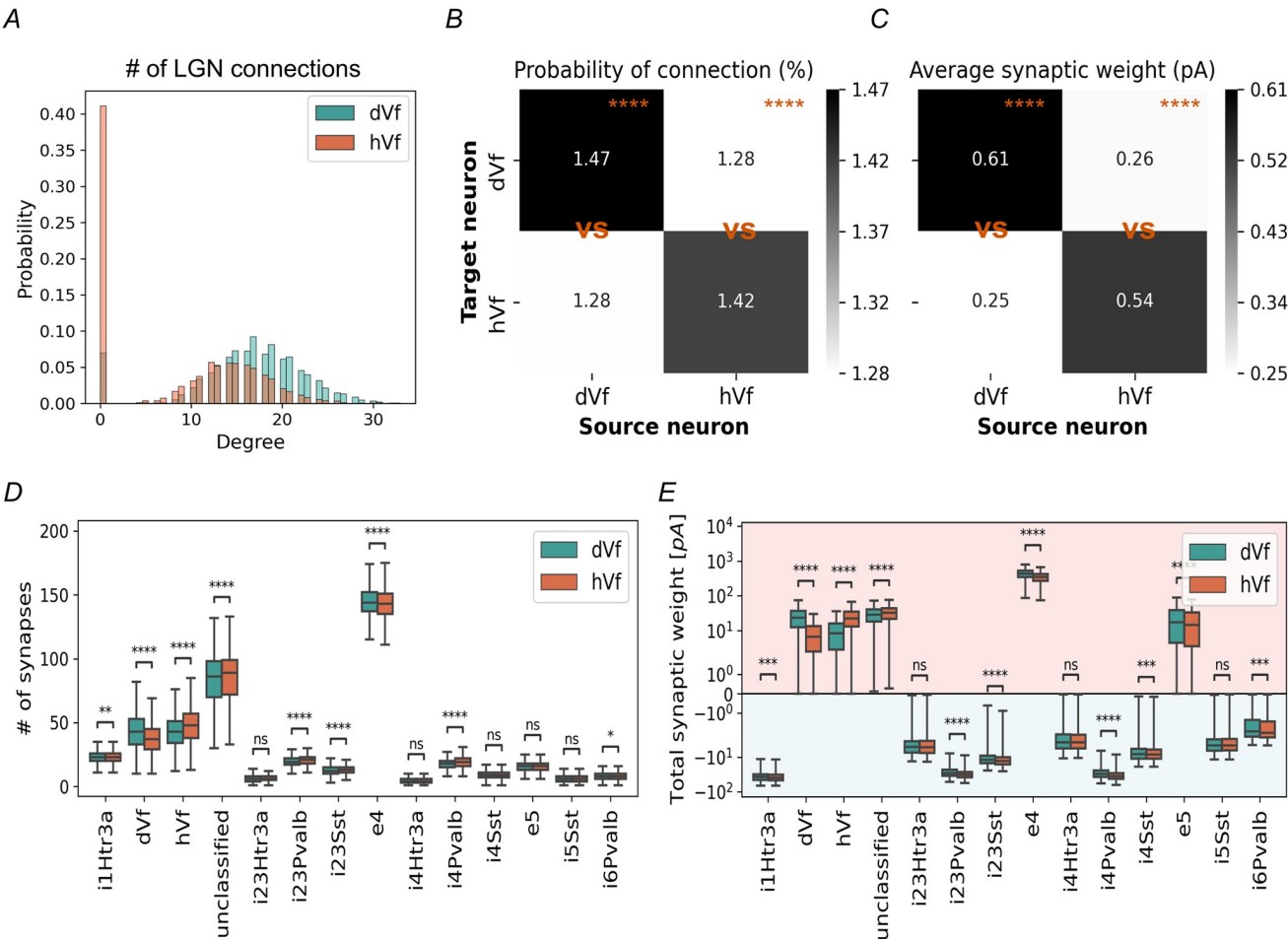

**Fig 4. Connectivity of excitatory L2/3 neurons.** (A) Distribution of in-degrees from LGN units for hVf (orange) and dVf (turquoise) neuron classes. (B) Probability of the connection (in %) between two randomly selected excitatory L2/3 neurons. Note the symmetry of the matrices, which shows that there is no bias in the source-target order. (dVf source, dVf vs hVf target: $t = 17.4$, $p = 2.7 \times 10^{-66}$; hVf source, dVf vs hVf target: $t = -14.1$, $p = 3.3 \times 10^{-44}$, $n = 6311$, $n_{dVf} = 2948$, $n_{hVf} = 3363$, Welch's t-test). (C) Average synaptic weight of a given connection between two randomly selected excitatory L2/3 neurons. (dVf source, dVf vs hVf target: $t = 45.1$, $p = 0$; hVf source, dVf vs hVf target: $t = -37.8$, $p = 0$, $n = 6311$, $n_{dVf} = 2948$, $n_{hVf} = 3363$, Welch's t-test). (D) Number of synapses with different populations of presynaptic neurons for the hVf (orange) and dVf (turquoise) classes. (E) Average synaptic weight with different populations of presynaptic neurons for the hVf (orange) and dVf (turquoise) classes. Shading indicates regions of excitatory (red) and inhibitory (blue) synaptic weights.

However, due to the relative scarcity of neurons in each excitatory L2/3 class and the relatively weak nature of their synapses, the total synaptic weight between these classes was negligible compared to inputs from excitatory L4 neurons (Fig 4D and 4E). Consequently, excitatory L2/3 neurons had a minimal direct influence on the specific responses of other excitatory L2/3 neurons.

Further investigation of the in-degree and out-degree, i.e. number of outgoing connections of a neuron, of both classes revealed a similarity in the recurrent V1 connections (S2 Table). Notably, dVf and hVf neurons exhibited similar distributions of presynaptic neuron types (Fig 4D). However, dVf neurons possess slightly stronger incoming recurrent synapses, primarily due to contributions from excitatory L4 neurons (Fig 4E), and increased LGN input (Fig 4A and S3 Table).

It is important to note that connectivity rules of the Billeh model (see Methods for details) may introduce some differences regarding the spatial distribution of dVf, hVf, and unclassified neurons. However, a spatial analysis reveals that all these excitatory L2/3 neurons are evenly distributed within the model's radius and depth (S4 Fig). Exploring spatial clustering through Ripley's *K* function [43] indicates that dVf and hVf neurons do not indicate strong preferences for grouping functionally similar neurons. While slight deviations from a random distribution are observed, there is insufficient evidence to conclude substantial proximity preferences between dVf and hVf neurons (S4 Fig).

## Dynamical origin of excitatory L2/3 classes

The role of network connectivity is evident in the emergence of dVf and hVf neurons. However, to gain a more complete understanding of the mechanism that segregates dVf and hVf neurons, it is necessary to analyze the dynamics of the network. To this end, we determined effective synaptic weights, which combine synaptic weight and presynaptic neuron activity for each pair of neuron types (see Methods for details) (Fig 5, S5 and S6 Figs). This comprehensive dynamic exploration of the main sources of recurrent current is essential to understand the mechanism leading to the segregation of the dVf and hVf neurons.

Notably, dVf neurons experienced a substantial increase in recurrent excitation (mainly from excitatory L4 neurons) and LGN inputs (Fig 5A). This recurrent excitation overcame the increased recurrent inhibition, mainly from L2/3 and L4 Pvalb populations, resulting in the depolarization of these neurons. On the contrary, the hVf neurons experienced a slight increase in excitation received from LGN and L4 neurons which was not sufficient to overcome inhibition originating primarily from L2/3 and L4 Pvalb interneurons, leading to hyperpolarization of these neurons (Fig 5A). Somewhere between these two classes of neurons were unclassified neurons, where inhibition originating from Pvalb interneurons was balanced by excitation originating from layer 4 excitatory neurons and LGN input (Fig 5B). This indicates that the different behavior of excitatory L2/3 neurons is the result of the interplay between LGN input excitation, excitation from L4 neurons, and inhibition from both L2/3 and L4 Pvalb interneurons.

Further insights arise from comparative analysis with experimental evidence in the primary somatosensory cortex, where highly active L2/3 excitatory neurons received enhanced excitatory inputs from layer 4, highlighting the relevance of network connectivity in shaping these neuron subsets [45].

Therefore, a comprehensive dynamic exploration of the main sources of recurrent current, namely L4 excitatory neurons, L4 Pvalb interneurons, and L4 Pvalb interneurons, is essential to understand the mechanism leading to the segregation of the dVf and hVf neurons.

**L4 excitatory neurons.**   Operating as amplifiers of LGN excitation (Fig 5B), excitatory L4 neurons received increased LGN bottom-up input, which triggered a positive feedback loop within the population. This result was consistent with the literature, positioning L4 excitatory neurons as upstream components in the specialized local excitatory network for sensory processing [20].

**L4 Pvalb interneurons.**   Similar to their excitatory counterparts, L4 Pvalb interneurons respond to elevated activity from LGN and L4 excitatory neurons, resulting in the increased inhibitory influence that is directed to the L2/3 circuitry (Fig 5C) [52].

**L2/3 Pvalb interneurons.**   Crucial to the emergence of dVf and hVf neurons, L2/3 Pvalb interneurons significantly depolarize during the visual flow onset (Fig 5C and S1 Fig). Activation of dVf neurons notably drives this depolarization, interacting dynamically with L2/3 Pvalb interneurons. Intriguingly, these inhibitory Pvalb neurons do not preferentially connect

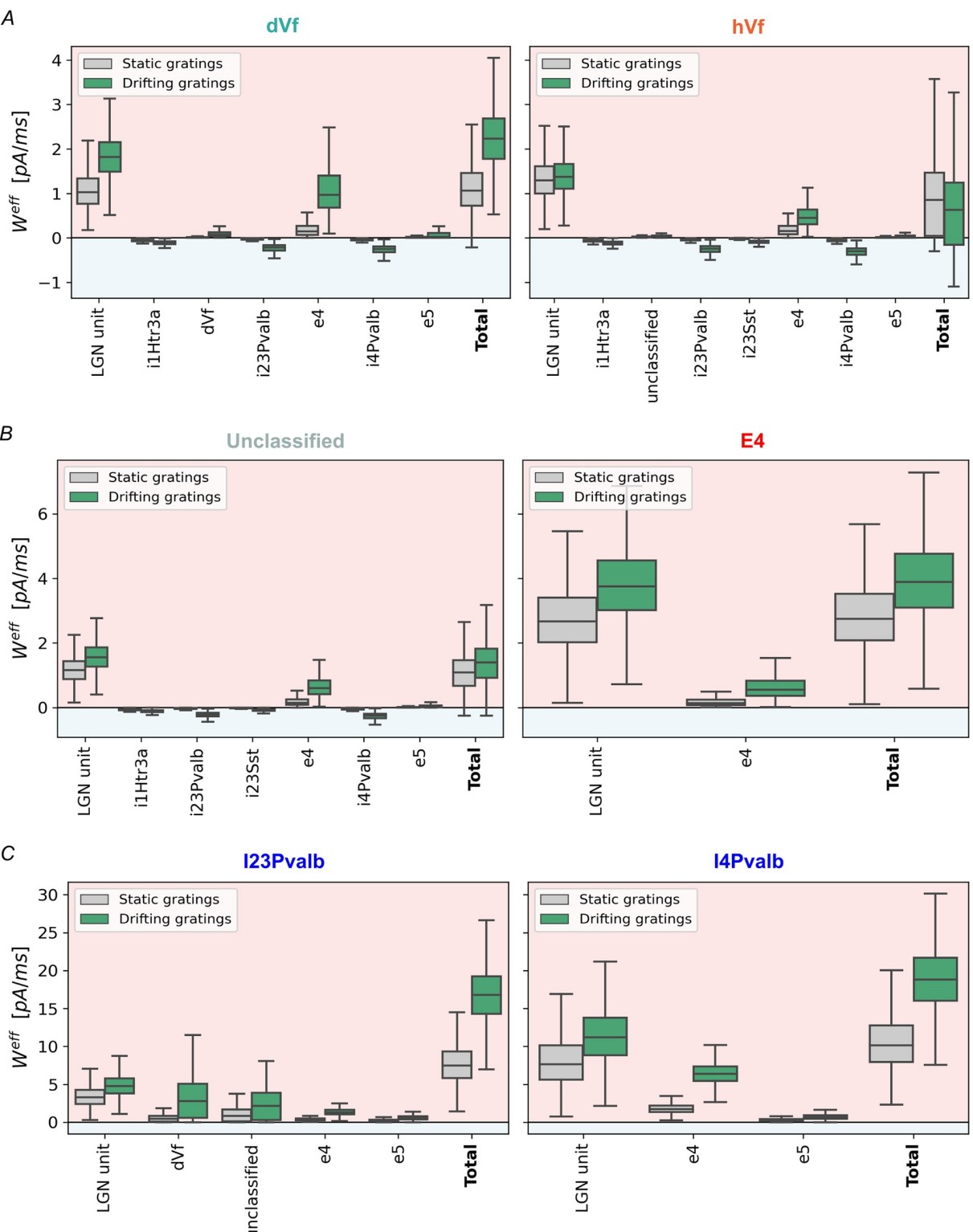

**Fig 5. Dynamic origins of excitatory L2/3 hyperpolarizing and depolarizing responses to the onset of visual flow.** (A) Effective synaptic input weight for dVf and hVf neurons. Bars represent the effective weight 500*ms* before (orange) and 500*ms* after (purple) the onset of visual flow. (B) As in (A), but for layer 2/3 unclassified neurons and layer 4 excitatory neurons. (C) As in (A), but for Pvalb neurons in layers 2/3 and 4. Notably, the comparison between static and drifting gratings' effective synaptic weights is statistically significant for all presynaptic populations, with p-values of 0.0001 or lower.

with any specific excitatory counterparts (dVf, hVf, or unclassified) (S7(A) Fig) when the total synaptic weight was divided by the number of presynaptic neurons, in accordance to experimental evidence [47].

Additionally, other L2/3 inhibitory neuron types (Htr3a and Sst), which depolarized as a result of the onset of visual flow (S6 Fig and S1 Table), have minor contributions to the dynamics of dVf and hVf neurons (Fig 5A). While experimental evidence suggests that Htr3a and Sst neurons are generally less excited by LGN inputs [46], with the influence of top-down inputs they could play a more substantial role in the formation of prediction error neurons, as suggested by computational models [21, 22].

Finally, our analysis revealed that depolarization of excitatory L5/6 neurons is triggered primarily by excitatory inputs of LGN and L4, along with recurrent intralayer excitation (S6 Fig). Hence, the lack of hyperpolarizing responses in L5/6 results from insufficient visually-driven recurrent inhibition, thus disabling the establishment of a multi-pathway excitatory-inhibitory balance (S1 Table).

## Impact of feature selectivity in excitatory L2/3 neuron classes

In light of the "like-to-like" connectivity principles employed in constructing the V1 model, it is possible that differences might exist in the direction tuning of dVf and hVf neurons [32]. The preferred direction of stimulus motion of each neuron, as assigned during model design, varied between the different excitatory L2/3 neuron classes (Fig 6A). For instance, dVf neurons exhibited a preference for vertical gratings drifting horizontally, aligning with the perturbation direction —a correlation validated through experimental findings [15]. In contrast, hVf and unclassified neurons exhibited a slight preference towards horizontally-oriented gratings undergoing vertical motion.

Previously, excitatory L2/3 neurons were categorized as dVf or hVf based on their response to horizontal drifts of vertical gratings at 2 Hz. Considering the sensitivity of dVf and hVf neurons to drifting gratings, it was plausible to hypothesize that hVf neurons might exhibit a preference for static gratings over drifting ones, and vice versa for dVf neurons, aligning with experimental observations where visual flow and running speed were uncoupled [15]. To further investigate this, the grating drift speed was increased to 8 Hz, leading to a reclassification of excitatory L2/3 neurons based on their response to this new stimulus. Interestingly, our findings revealed that neuron assignments to dVf and hVf classes remained largely unaffected, with the PSC responses maintaining a comparable magnitude (Fig 6B and 6C). This outcome stands in contrast to the temporal frequency preferences exhibited by perturbation-sensitive neurons [15], as well as the behaviour of mismatch-responsive neurons, which demonstrated linear scaling of responses with the difference between running speed and visual flow speed [10, 53]. This observation highlights a specific limitation within the Billeh model when handling different temporal dynamics of visual stimuli.

Our analysis revealed that dVf neurons exhibited a particular preference for the vertical orientation of the gratings (Fig 6A). Could this mean that the particular behavior of a dVf or hVf neuron depended exclusively on the orientation of the stimulus, unveiling them as purely orientation-selective neurons? To tackle this question, we considered a visual stimulus consisting of horizontal gratings drifting vertically at 2 Hz, to explore the effect of orientation tuning. In this scenario, the reclassification of excitatory L2/3 neurons diverged significantly from the initial classification, implying a strong influence of grating orientation on the individual behavior of these neurons. This result was expected since the Billeh model supports orientation selectivity.

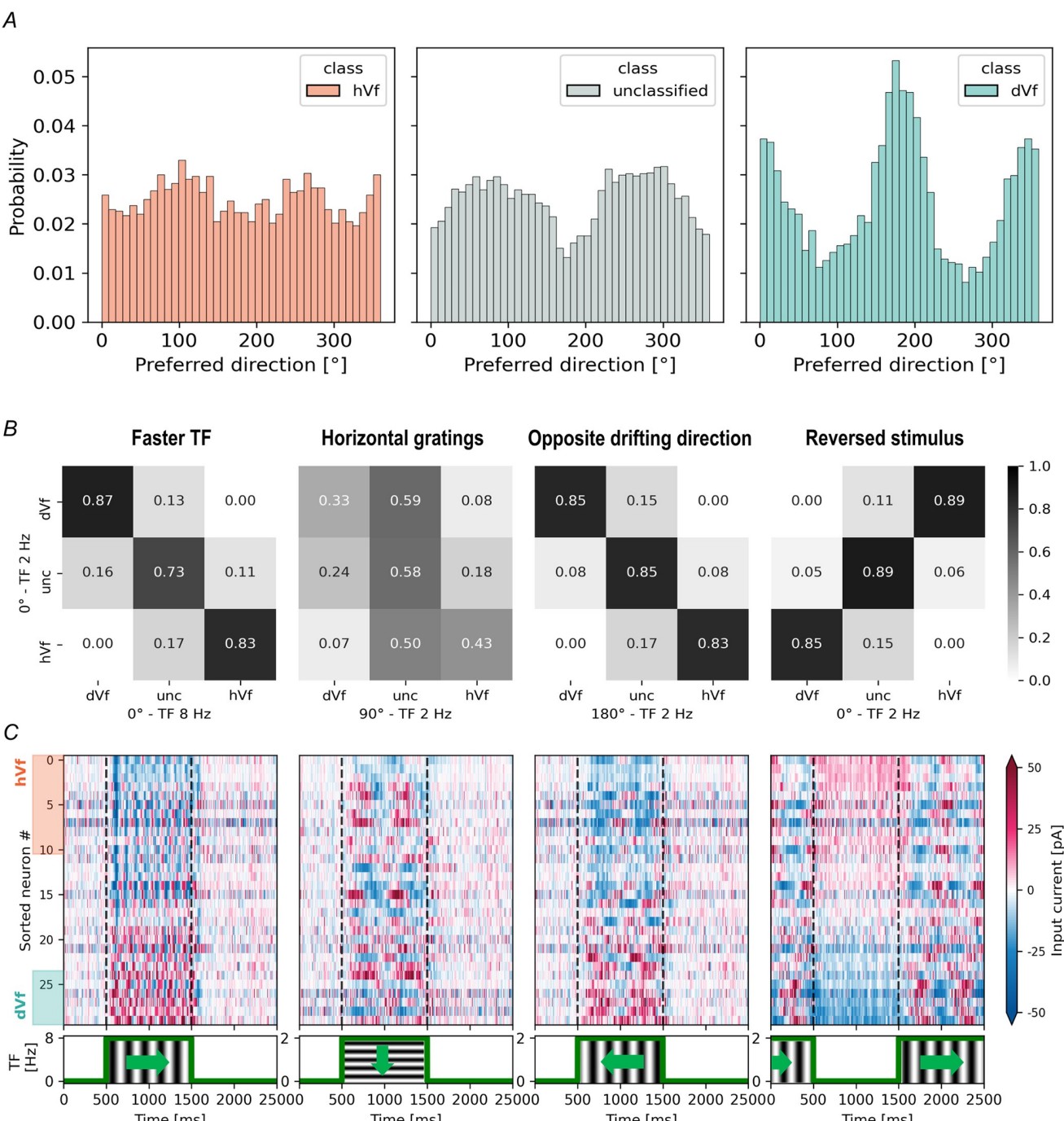

**Fig 6. Responses of excitatory L2/3 neuron classes to various visual input features.** (A) Distribution of tuning angles for visual flow among excitatory L2/3 hVf (left), unclassified (middle), and dVf (right) neurons. (B-C) The different visual stimuli consisted of a faster visual flow composed of vertical gratings moving at 8Hz (left), horizontal gratings with vertical motion (middle left), reversed direction of motion (middle right), and a sudden visual flow halt (right). (B) Confusion matrices showing the fraction (0–1) of excitatory L2/3 neurons classified in each class according to their response to the novel visual stimulus (horizontal axis) compared to the classification from the original experiment (vertical axis). (C) Heatmap representing input current responses to the given stimulus for the same neurons depicted in Fig 3A. Here, TF denotes the temporal frequency of the visual flow. Note the different y-axis scale in left stimulus panel of (C).

We also analyzed the effect of reversing the motion direction of vertical gratings drifting at 2 Hz (Fig 6B and 6C). Notably, the input currents displayed minimal changes compared to the original visual stimulus, and the reclassification of excitatory L2/3 neurons demonstrated a high correlation with the original classification. Consequently, the dynamic behavior of these neurons appears unaffected by the visual flow direction.

Finally, we subjected the Billeh model to a sudden halt of the visual flow. As expected, the behavior of dVf and hVf neurons was reversed: hVf neurons depolarized and dVf neurons hyperpolarized as a response to the visual flow perturbation (Fig 6B and 6C), similar to what was found experimentally [11]. Importantly, our observations occurred without top-down input influence.

## Direction selectivity vs Perturbation sensitivity

Considering the relevance of grating orientation in the emergence of the dVf and hVf neuron classes, an intriguing question arose: Could there be neurons exhibiting consistent behavior regardless of the direction of grating motion? An affirmative response to this query would suggest that these neurons primarily react to the perturbations of the visual flow. This phenomenon has already been observed experimentally in V1 neurons [15].

Previously, excitatory L2/3 neurons had been classified as dVf or hVf based on their responses to horizontally drifting gratings. In this study, these neurons were reclassified based on their average input current responses to visual flow onset across a wide range of visual flow directions ($0° : 315° : 45°$). Intriguingly, a subset of dVf (241/2948) and hVf (519/3363) neurons exhibited consistent responses across all drifting directions, i.e., this subset of dVf/hVf neurons depolarized/hyperpolarized for every gratings direction (S8 Fig). These neurons were categorized as **perturbation-sensitive neurons**, as they exhibited significant responses to sudden changes in the visual flow, regardless of its direction. Remarkably, the response magnitude remained fairly uniform across visual flow directions, with a slight preference for the front-to-back direction (S8 Fig), aligning with experimental observations [54]. These perturbation-sensitive neurons emerged as suitable candidates for encoding sudden visual flow changes in open-loop scenarios, aligning with their potential role as prediction error neurons in such contexts. Particularly, dVf and hVf neurons are well-positioned to function as positive and negative prediction error neurons, respectively, under these circumstances.

The consistent responses of perturbation-sensitive neurons may be stem from their intrinsic preferences. Specifically, it is plausible that hVf neurons tend to favor slower visual flow speeds, such as static gratings, while dVf neurons lean towards faster visual flow speeds, indicating a positive response to increased speed. To investigate this, we examined the responses of these neurons across various visual flow temporal frequencies ($1 : 9 : 1$ Hz) while maintaining a vertical grating orientation. The preferred frequency for each neuron in this orientation was determined as the frequency at which the dVf/hVf neuron received the highest/lowest input current during visual flow. Remarkably, both dVf and hVf neurons exhibited similar distributions of preferred frequencies (S8 Fig), peaking around 5 Hz.

Conversely, a subset of dVf (892/2948) and hVf (503/3363) neurons exclusively responded to visual flow onset in a particular direction while showing unclassified behavior in others, so we identified them as **direction-selective** neurons.

To conclusively determine if perturbation-sensitive neurons are indeed encoding visual perturbations, i.e. sudden changes in visual input, we presented the model with black-to-white full-field flash and tracked the responses of excitatory L2/3 neurons (Fig 7 and S9 Fig). Notably, dVf/hVf perturbation-sensitive neurons exhibited an abrupt increase/decrease in input current in response to the flash, regardless of whether the transition was from black-to-white

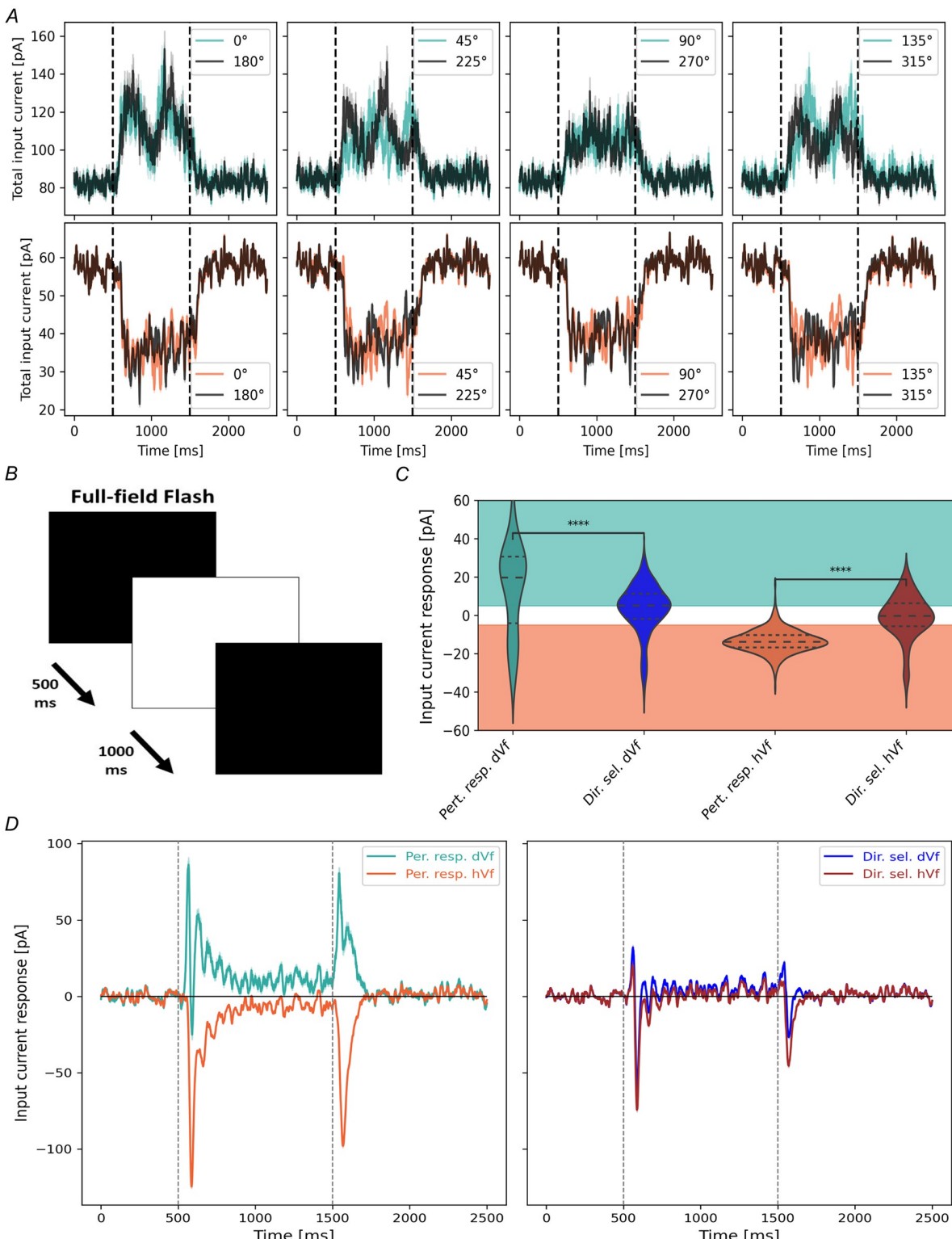

**Fig 7. A subset of excitatory L2/3 neurons exhibit perturbation sensitivity.** (A) Input current traces for perturbation-sensitive dVf (top) and hVf (bottom) neurons in response to various grating directions. (B) Schematic of the full-field flash visual stimulus. (C) Average input current responses to the full-field flash among distinct subsets of excitatory L2/3 neurons (Dir. sel. dVf vs Pert. resp. dVf, $t = 6.78$, $p = 7.3 \times 10^{-11}$; Dir. sel. hVf vs Pert. resp. hVf, $t = -22.1$, $p = 1.2 \times 10^{-83}$, Welch's t-test). Turquoise and orange shading represent dVf and hVf classification regions. (D) Input current responses of perturbation-sensitive and direction-selective excitatory L2/3 neurons to the full-field flash. Vertical dotted lines mark the full-field flash. Shaded areas indicate SEM.

or from white-to-black (Fig 7C and 7D). This response faded after a few hundred milliseconds, with the neuron's postysnaptic currents returning to baseline levels. In contrast, direction-selective neurons demonstrated a significantly different response, with both dVf and hVf direction-selective neurons exhibiting similar behavior (Fig 7C and 7D). These findings confirmed that perturbation-sensitive neurons acted as broad detectors of visual input perturbations, while direction-selective neurons focused primarily on detecting the movement in a particular direction.

## Discussion

The emergence of different neural responses to visual flow perturbations in the primary visual cortex (V1) has drawn attention due to its relevance to sensory perception, attentional modulation, and decision-making [4, 6, 11, 15]. The discovery of error neurons in L2/3 that report positive and negative errors between visual flow prediction based on motor signals and actual visual flow has opened avenues for investigating the neural mechanisms underlying prediction and adaptation [11, 55]. Notably, subsequent work showed that similar neural responses can also be detected even in situations where animals remain stationary, suggesting that these error neurons rather report changes in visual flow [15].

This study aimed to explore the emergence of distinct perturbation-responsive V1 neuron classes in the absence of a sensorimotor expectation using the Billeh model, a biologically realistic computational model [32]. Our investigation revealed the presence of two functional classes of excitatory L2/3 neurons, termed dVf and hVf (Fig 3), mirroring the depolarizing (dMM) and hyperpolarizing (hMM) mismatch neurons observed in previous experiments where visual flow was uncoupled from locomotion speed and visual flow perturbations appeared as sudden onsets of drifting gratings [11]. This unique behavior emerged despite that, in terms of model composition and wiring, the Billeh model assumes only one model of excitatory L2/3 neuron, showcasing the importance of the network dynamics in neural responses.

In particular, these two classes of neurons emerged due to differences in both synaptic and effective connectivity, particularly with inhibitory neurons. The relevance of connectivity in this context is not surprising, as theoretical studies have suggested that the connectome encodes essential features of the world within its weight distribution, enabling rapid adaptation to sensory evidence [56].

Divergent synaptic input patterns from the lateral geniculate nucleus (LGN) and excitatory L4 neurons led to varying levels of excitation, where dVf neurons exhibited stronger excitation and hVf neurons showed weaker visually-driven inputs (Figs 4 and 5A). Indeed, the LGN input did not have a direct functional influence on most hVf neurons (Fig 4A). This structural discrepancy profoundly shapes the distinct depolarizing and hyperpolarizing responses observed, presenting a testable prediction that could be experimentally validated if these two classes of neurons were identified.

Recent experimental findings suggested that positive and negative prediction error neurons in L2/3 map to different transcriptomically defined neuron types [57]. If that is correct, then one can expect the above-mentioned connectivity differences between these classes—namely, prominent differences in inputs to the different classes of excitatory L2/3 neurons from, e.g., L4, but relatively similar recurrent connections within L2/3. In this case, the genetic labels of these different classes can also help experimentalists to test our modeling predictions regarding connectivity.

Further dynamic analysis of the network revealed that dVf neurons influenced inhibitory Parvalbumin (Pvalb) neurons, indirectly inhibiting hVf neurons through a competitive interaction mediated by shared inhibition (Fig 8). Thus, this inhibitory connections from Parvalbumin neurons in layers 2/3 and 5 link excitatory L2/3 neurons across the fine-scale dVf and hVf

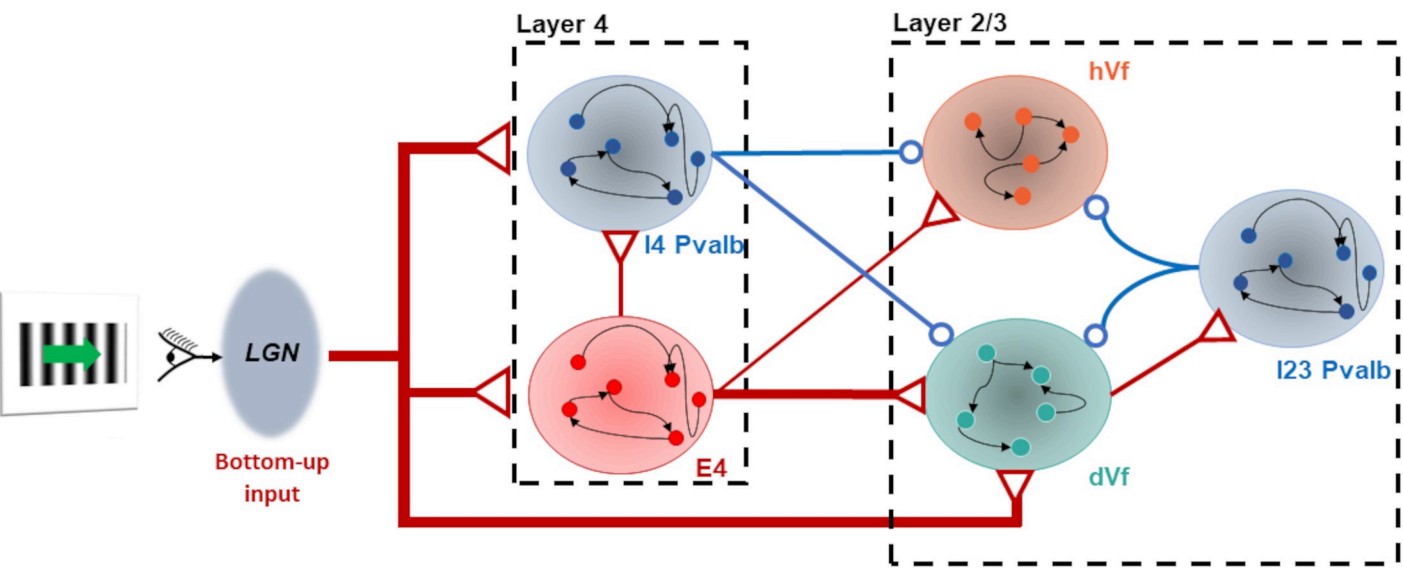

**Fig 8. Network dynamics reveal crucial L2/3 inhibitory pathways.** Diagram illustrating the influence of LGN bottom-up input on the segregation of hVf and dVf neurons. The arrows represent variations in effective synaptic weights triggered by the visual flow onset. Excitatory projections are denoted by red arrows, while inhibitory projections are indicated by blue arrows. The diagram selectively displays neuronal populations deemed pivotal in the functional division among excitatory L2/3 neurons.

subnetworks, in agreement with experimental studies [47]. Overall, this increased inhibition could not be overcome by the excitatory inputs received by hVf neurons, resulting in a visually driven hyperpolarization. On the other hand, unclassified neurons exhibited a balance between recurrent inhibition and excitation, resulting in minimal responses to visual flow perturbations (S1 Table) This interplay highlighted the significance of dynamic inhibitory circuits in shaping neuron responses.

Recent publications on cortical microcircuit modeling have revealed interesting conditions for the emergence of prediction error neurons [21, 22]. They considered a particular circuit of bottom-up and top-down connections and emphasized that, while the distribution of bottom-up and top-down inputs to excitatory cells was well studied, the distribution between different types of cortical inhibitory interneurons was less understood and probably diverse. Therefore, various input configurations led to different circuits for the prediction error neurons. In our study, the thalamocortical connectivity of the Billeh model, which is supported by experimental studies [46, 58, 59], suggests that the bottom-up input almost exclusively targets excitatory and Pvalb neurons in all layers and Htr3a neurons in L1. This led us to the circuitry shown in Fig 8, where dVf neurons indirectly inhibited hVf neurons during the onset of visual flow, by activating L2/3 Pvalb interneurons. This circuit is reminiscent of a *competition by common inhibition* between dVf and hVf neurons, where the former use the pool of Parvalbumin neurons to suppress the activity of the latter [60]. Further evidence for this hypothesis can be seen in Fig 3B, where the firing rate of dVf neurons is anti-correlated with the firing rate of hVf neurons. This outlines a modeling prediction of a connectivity motif, which our work suggests to be important for maintaining the dVf and hVf functional classes, and which can potentially be observed experimentally if one can correlate functional properties of L2/3 neurons with their local connectivity.

The study also delved into the role of visual input characteristics in shaping neuronal behaviors. In particular, it revealed that the identified dVf neurons tended to prefer the

orientation of the given grating relative to others, while the unclassified neurons tended to prefer perpendicular gratings (Fig 6A). Furthermore, when the vertical orientation of the gratings was changed to a horizontal orientation (Fig 6), some dVf and hVf neurons behaved as unclassified neurons, while others continued to behave in the same way. Furthermore, a subset of dVf and hVf neurons exhibited consistent responses across different directions, indicative of their role in detecting general visual input perturbations. This was confirmed through a full field flash visual stimulus (Fig 7). Consequently, this subset of dVf and hVf neurons aligned with the positive and negative perturbation-sensitive neurons found in previous experiments [15].

It is important to acknowledge the role of additional factors, such as locomotion-related inputs [14, 16–19, 55] and feedback from higher visual areas [61], in modulating V1 neuron responses to visual flow perturbations. Particularly, locomotion has been shown not only to modulate V1 activity [16], but also to strongly drive motor-related responses in V1, even during dark running conditions [14]. The precise role of locomotion-induced top-down input remains a subject of ongoing investigation; it potentially operates as a gain modulator, enhancing V1 neuron responses while preserving orientation selectivity [15, 16]. This amplification effect appears to be smoothly contingent on the animal's speed [18]. Alternatively, this input may also provide a prediction of incoming visual flow which is dependent on previous visuomotor coupling experience, leading to the emergence of prediction error neurons [11, 14, 55]. Recent experimental evidence suggests that mouse visual areas prioritize the encoding of potentially fast-changing and behaviorally relevant visual features during locomotion [44], which may imply that a larger number of perturbation-sensitive neurons would appear in the presence of locomotion. Considering these insights alongside the pronounced preference of dVf neurons for gratings aligned with their preferred direction, it becomes plausible that locomotion amplifies the number of perturbation-sensitive neurons and accentuates the differentiation between dVf and hVf neurons. In general, these findings suggest that top-down locomotion inputs play a crucial role in modulating V1 neuron responses to visual flow perturbations, enabling efficient encoding and integration of behaviorally relevant visual inputs.

Finally, it is worth mentioning that, although the version of the Billeh model that we used systematically integrates a wide range of detailed biological data, it describes neuronal cells as point neurons. Recent modeling and experimental studies suggest that dendritic arbours may play a signification role in the computation of prediction errors [21, 61, 62]. Hence, studying the roles of dendritic computations in producing the effects we described above would be a potentially fruitful area for future work.

To conclude, the insights derived from the Billeh model provide a window into the dynamics of V1 neural responses to visual flow perturbations, highlighting the complex interplay between synaptic connectivity, inhibitory circuits, and dynamic inputs. The emergence of distinct response classes, in the absence of explicit sensorimotor expectations, underscores the computational richness of V1's microcircuitry.

## Supporting information

**S1 Fig. Illustration of the procedure used to determine the rheobase of cell models.** Top: Injected current into the cell soma consisting of increasingly larger current steps interleaved by resting periods. The red line represents the identified rheobase. Middle: Cell model membrane voltage as a response to the injected current. The red line represents the voltage threshold in the model. Bottom: After-spike currents of the model. Sharp vertical lines indicate the presence of a spike.
(TIF)

**S2 Fig. Responses to sudden onset of visual flow for representative excitatory L2/3 neurons.** (A) Left: Neuron's membrane voltage response (top), input current response (middle), and firing rate (bottom) for a sample dVf neuron. The mean input current during visual flow (blue horizontal line) and the classification thresholds (black dots) are shown. Right: Responses of the different PSC sources for a sample dVf neuron, including total input current (red), recurrent current (olive), and LGN current (blue). Vertical dashed lines mark the visual flow period. (B) As in (A), but for a sample unclassified neuron. (C) As in (A), but for a sample hVf neuron.
(TIF)

**S3 Fig. Baseline distributions of excitatory L2/3 neurons.** (A) Distribution of baseline input current mean for excitatory L2/3 classes (ANOVA: $F = 162.8$, $p = 1.5 \times 10^{-70}$, $n = 12689$; dVf vs hVf: $t = 15.72$, $p = 1.5 \times 10^{-54}$, $n = 6311$; dVf vs unc: $t = 4.18$, $p = 2.9 \times 10^{-5}$, $n = 9326$; hVf vs unc: $t = 13.6$, $p = 2.8 \times 10^{-41}$, $n = 9741$, Welch's t test). (B) Distribution of baseline input current standard deviation (SD) for excitatory L2/3 classes (ANOVA: $F = 14.96$, $p = 3.2 \times 10^{-7}$, $n = 12689$; dVf vs hVf: $t = 5.24$, $p = 1.7 \times 10^{-7}$, $n = 6311$; dVf vs unc: $t = 3.58$, $p = 3.5 \times 10^{-4}$, $n = 9326$; hVf vs unc: $t = 2.72$, $p = 0.0065$, $n = 9741$, Welch's t-test). Red triangles indicate the class mean.
(TIF)

**S4 Fig. Spatial distribution of dVf, hVf, and unclassified neurons.** (A) The density distribution of the distance to the center of the column, which we refer to as radius, (left), and the depth within the layer (right) is depicted for dVf, hVf, and unclassified neurons. (B) Ripley's $K$ divided by the layer 2/3 area as a function of the search radius $t$. The colors of the curves and dots represent the results for dVf (turquoise), hVf (orange), and unclassified (gray) neurons. The black points represent the outcome of a random null model of 10,000 neurons averaged over 100 realizations. Also, the excitatory L2/3 classes are represented in the V1 cylinder.
(TIF)

**S5 Fig. Full dynamic origins of excitatory L2/3 hyperpolarizing and depolarizing responses to the onset of visual flow.** Effective synaptic weight for various neuron types: (A) dVf and hVf neurons; (B) L2/3 unclassified and L4 excitatory neurons; and (C) L2/3 and L4 inhibitory Pvalb neurons. The bars represent the effective weight $500ms$ before (orange) and $500ms$ after (purple) the onset of visual flow.
(TIF)

**S6 Fig. Distributions of effective synaptic weights for different neuronal populations.** Effective synaptic weight for various neuron types: (A) L1 and L2/3 inhibitory Htr3a neurons; (B) L2/3 and L4 inhibitory Sst neurons; and (C) excitatory L5 and L6 neurons. The bars represent the effective weight $500ms$ before (orange) and $500ms$ after (purple) the onset of visual flow.
(TIF)

**S7 Fig. Presynaptic weight distributions for different neuronal populations.** Total synaptic weight of several presynaptic populations for: (A) L2/3 and L4 inhibitory Parvalbumin neurons; (B) L2/3 unclassified and excitatory L4 neurons; and (C) excitatory L5 and L6 neurons.
(TIF)

**S8 Fig. Orientation and frequency preferences of excitatory L2/3 perturbation-responsive neurons.** (A) Input current responses in various drift directions normalized to the input current responses to horizontal drift (0° direction). (B) Distribution of preferred visual flow frequency of dVf (turquoise) and hVf (orange) neurons, and horizontal drift vertical gratings.
(TIF)

**S9 Fig. Raster plot of V1 model response to the full-field flash stimulus.** Top: Raster plot of the spike response of LGN units to full-field flash. Bottom: Laminar raster plot of the spike response of V1 neurons to full-field flash. The colors of the spikes represent the different populations of neurons, following the same palette as in Fig 1. Vertical dashed lines indicate the period of full-field flash.
(TIF)

**S1 Table. Input currents for different neuron populations.** Mean values for the different PSC sources during the static gratings (baseline) and drifting gratings (visual flow) periods. In some populations, a classification of the belonging neuron's response to the drifting gratings was made (see Methods for details). The SEM on the sample population and realizations is taken as the error. The total current considers the contributions of the recurrent, bottom-up, ASC and BKG currents.
(PDF)

**S2 Table. In/Out degrees for excitatory L2/3 classes.** Connections with other V1 neurons (recurrent), thalamus (LGN), and noisy background sources (BKG) are considered. The SEM for each variable is taken as the error.
(PDF)

**S3 Table. Weighted in/out degrees for excitatory L2/3 classes.** Weighted in/out-degrees, i.e. the number of incoming/outgoing connections multiplied by the synaptic weight, with other V1 neurons (recurrent), thalamus (LGN), and noisy background sources (BKG) are considered. The SEM for each variable is taken as the error.
(PDF)

**S4 Table. Statistical significance testing results for the comparison of total presynaptic weight sources between dVf and hVf neurons.** The table presents the statistical significance testing results, providing insights into the microcircuitry differences between dVf and hVf neurons. Welch's t-test were performed, with the low p-values attributed to the large number of neurons within both dVf and hVf classes.
(PDF)

**S5 Table. Statistical significance testing results for comparing the number of synapses by source type between dVf and hVf neurons.** The table presents the statistical significance testing results, providing insights into the microcircuitry differences between dVf and hVf neurons. Welch's t-test were performed, with the low p-values attributed to the large number of neurons within both dVf and hVf classes.
(PDF)

## Acknowledgments

We thank the Allen Institute founder, Paul G. Allen, for his vision, encouragement, and support.

## Author Contributions

**Conceptualization:** José J. Ramasco, Anton Arkhipov, Wolfgang Maass, Claudio R. Mirasso.

**Formal analysis:** J. Galván Fraile.

**Investigation:** Anton Arkhipov, Wolfgang Maass, Claudio R. Mirasso.

**Project administration:** Claudio R. Mirasso.

**Software:** J. Galván Fraile, Franz Scherr.

**Supervision:** Anton Arkhipov, Wolfgang Maass.

**Writing – original draft:** J. Galván Fraile.

**Writing – review & editing:** J. Galván Fraile, José J. Ramasco, Anton Arkhipov, Wolfgang Maass, Claudio R. Mirasso.

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
