## [Decision Letter · Decision Letter 0]

12 Dec 2023

Dear Prof. Mirasso,

Thank you very much for submitting your manuscript "Modeling circuit mechanisms of opposing cortical responses to visual flow perturbations" for consideration at PLOS Computational Biology. As with all papers reviewed by the journal, your manuscript was reviewed by members of the editorial board and by several independent reviewers. The reviewers appreciated the attention to an important topic. Based on the reviews, we are likely to accept this manuscript for publication, providing that you modify the manuscript according to the review recommendations.

Sincerely,

Lyle Graham

Section Editor

PLOS Computational Biology

Reviewer's Responses to Questions

**Comments to the Authors:**

Reviewer #1: The authors have studied the circuit and dynamical mechanisms underlying the different responses of the primary visual cortex to visual stimuli. Using a biologically realistic model of V1, the authors have tried to explain how the incoming excitatory stimuli from LGN can depolarize some excitatory neurons in layer 2/3 while hyperpolarizing the other ones. The authors have shown that how differential excitatory inputs from LGN combined with the recurrent inhibition give rise to this observation. The structural and dynamical aspects of the problem have been extensively studied.

I found the manuscript well-written and clear and the results are interesting. Using a detailed biological model is an advantage of the study and can help for more future researches using these type of models. I can recommend publication of the manuscript after addressing some points that I list below. Most of all, I think the Methods section needs a revision with taking more care.

Figure 1: The quality of the figure should be improved. Text in A cannot be read.

Model is general, is a little bit vague. It seems that it is better to first introduce general model including second part of Eq. 1 and then Eq. 2. The first part of Eq. 1 (ASC) should be defined after Eq 2. Furthermore, ACS should be explained in more details since it is not known for the most people in the field and is not exactly explained.

Eq. 3 also looks imprecise. In the equations, t should be the time of the firing of the presynaptic neurons. Therefore, it seems more precise that instead of time t, write it in the form of (t-t_sp) where t_sp is the time of firing of the presynaptic neuron or external input.

Again, in Eq. 5 I_e(t) is written in another form. I think there is some discrepancy in writing the methods. I understand what the authors meant, but for being comprehensible for general reader and for being reproducible a more coherent and precise format in needed.

Fig. 3: In the caption please note what are different columns in part A.

Line 325: “ hVf neurons lack direct connections to the LGN” should it be from LGN?

Line 326: “This discovery aligns with” might be “this observation aligns with”.

Lines 332-337: I do not agree that this is a rich-club structure. Indeed this is more looks like a modular structure. In rich-club structure there is a subset of the nodes in different modules that are more strongly connected to each other. Please write this part with more care.

Lines 349-355: I would suggest moving this part to a relevant position in Discussion section.

From line 391: appearance of these three subtitles here without any prior explanations is not appropriate. Please write and explanations before them why they are chosen for more detailed analysis.

Reviewer #2: uploaded as attachmentt

**Have the authors made all data and (if applicable) computational code underlying the findings in their manuscript fully available?**

Reviewer #1: None

Reviewer #2: **No: **On the manuscript it states: Data and code will be accessible via gitHub

Code was not available during the review process

PLOS authors have the option to publish the peer review history of their article (what does this mean?). If published, this will include your full peer review and any attached files.

Reviewer #1: **Yes: **Alireza Valizadeh

Reviewer #2: No

Figure Files:

Data Requirements:

Reproducibility:

References:

---

## [Decision Letter · Decision Letter 1]

18 Feb 2024

Dear Prof. Mirasso,

We are pleased to inform you that your manuscript 'Modeling circuit mechanisms of opposing cortical responses to visual flow perturbations' has been provisionally accepted for publication in PLOS Computational Biology.

Best regards,

Lyle J. Graham

Section Editor

PLOS Computational Biology

Reviewer's Responses to Questions

**Comments to the Authors:**

Reviewer #1: I have no further comment.

Reviewer #2: The authors have addressed all my concerns satisfactorily. I believe the manuscript is suitable for publication in its current form. I thank the authors for this useful research.

**Have the authors made all data and (if applicable) computational code underlying the findings in their manuscript fully available?**

Reviewer #1: Yes

Reviewer #2: Yes

PLOS authors have the option to publish the peer review history of their article (what does this mean?). If published, this will include your full peer review and any attached files.

Reviewer #1: **Yes: **Alireza Valizadeh

Reviewer #2: No

---

## [Editor Report · Acceptance letter]

4 Mar 2024

PCOMPBIOL-D-23-01618R1 

Modeling circuit mechanisms of opposing cortical responses to visual flow perturbations

Dear Dr Mirasso,

I am pleased to inform you that your manuscript has been formally accepted for publication in PLOS Computational Biology. Your manuscript is now with our production department and you will be notified of the publication date in due course.

With kind regards,

Anita Estes
